# The participation of Japan in the EISCAT Scientific Association

Nobuo Matuura[1], Ryoichi Fujii[2], Satonori Nozawa[3]

[1]Professor Emeritus of Nagoya University

[2]Research Organization of Information and Systems, National Institute of Polar Research, 10-3, Midori-cho, Tachikawa-shi, Tokyo 190-8581, Japan

[3]Institute for Space-Earth Environmental Research, Nagoya University, Furo-cho F3-3, Chikusa-ku, Nagoya, 464-8601, Japan

*Correspondence to*: Ryoichi Fujii (eiscatfujii@gmail.com)

**Abstract.** In Chapter 1, the original planning of Japanese Svalbard IS Radar with phased array antennas is described. In 1988 the plan was proposed as one of major projects for the forthcoming Solar-Terrestrial Environment Laboratory, Nagoya University to be reorganized from the Research Institute of Atmospherics in Nagoya University. On the other hand, in 1989 UK scientists proposed a plan of the Polar Cap Radar with parabolic dish antennas in Longyearbyen to the EISCAT Council. In Chapter 2, the circumstances arriving in the Japan's participation in the EISCAT Scientific Association with details of its processes with strong collaborations with Norwegian scientists and the EISCAT Scientific Association are described. In 1995 Japan participated EISCAT Scientific Association as the seventh member country with funds for contribution to the second dish antenna of the EISCAT Svalbard Radar. In Chapter 3, a summary of the EISCAT related achievement by Japanese scientists is described, where major interests are the lower thermosphere wind dynamics, the Magnetosphere-Ionosphere-Thermosphere coupling, characteristics and driving mechanisms of ion upflow, electrodynamics of current, electric field and particles, characteristics and production mechanisms of auroras such as pulsating aurora, and aurora tomography. In Chapter 4, summary of scientific collaborations between Japan and Europe, particularly, those between Japan and Norway, and hope for the forthcoming EISCAT_3D and further collaboration with EISCAT community are described.

## 1 Original Planning of Svalbard Incoherent Scatter Radar (SIR) in Japan

In mid80' the Ministry of Education, Science, Sports and Culture (hereinafter abbreviated as "MEXT"), Japan, appointed the Research Institute of Atmospherics, Nagoya University to be reorganized. The Research Institute of Atmospherics had been established in 1949 and it aimed to research atmospherics and related natural phenomena such as VLF electromagnetic waves. In 1987 soon after this appointment Professor Sachio Hayakawa, the President of Nagoya University, asked an academic community in Japan, the Society of Geomagnetism and Earth, Planetary and Space Sciences, to discuss and make a plan for the reorganization. The Society appointed Professor Takasi Oguti, Geophysics Research Laboratory at University of Tokyo, to chair the reorganization committee that then made a plan to reorganize the Research Institute of Atmospherics into the

Solar-Terrestrial Environment Laboratory (STEL). The aim of the new laboratory was to study on the structure and dynamics of the solar-terrestrial system.

The committee represented by Professor Oguti set up four scientific plans as its major projects of the forthcoming STEL, one of which was a project to construct a Svalbard Incoherent Scatter Radar (hereinafter abbreviated as 'SIR') on its own. The motivation behind this project came from the fact that Svalbard was located statistically beneath the cusp, where solar wind plasmas and energy directly entered the magnetosphere and hence it was one of the key regions for the solar wind-magnetosphere-ionosphere interactions. A great advantage of Svalbard was that it was the only place beneath the cusp on the globe to have dark sky time in the midday during winter, which made it possible to observe cusp auroras. In order to prepare the SIR project, Professor Oguti asked Dr. Nobuo Matuura, the Communication Research Laboratory, the Ministry of Posts and Telecommunications, to join the STEL in 1988. Since then Professor Matuura was in charge of the SIR project together with Dr. Satonori Nozawa since 1989 and Dr. Ryoichi Fujii since 1992. The outline of the SIR was that the SIR would be a bi-static IS radar system with a transmitter/receiver dish antenna in Longyearbyen and a phased array, multiple beams receiving antenna in Ny Ålesund as shown in Fig. 1 (Matuura and Nozawa, 1991). The phased array radar was designed also to have the capability to transmit radar beams and to work as a mono-static radar. The planned radar frequency was 400-500 MHz and the peak powers of the transmitters were 3MW for the dish antenna (MSDC klystrons, TV klystrons) and 3-5 MW for the phased array antenna with 3000-5000 crossed dipoles (solid state module, 1kW each), respectively as shown in Fig. 2 (Matuura et al., 1990). It is noted that Kyoto University had successfully developed and installed a phased array IS radar system called Middle and Upper Atmosphere (MU) radar (Fukao et. al, 1985a, b) at Shigaraki in Japan during the Middle Atmosphere Project (MAP, 1982-1985) and they had been running it with obtaining noble, important atmospheric data that could not have been obtained before.

In September 1988, Professor Oguti who later became the first director of the STEL had visited Professor Asgeir Brekke and his colleagues at the Auroral Observatory of the University of Tromsø and asked Norwegian scientists to get together with Japan for the SIR project that would be independent of the EISCAT Scientific Association. On the other hand, the UK report "The Polar Cap Radar" signed by A. P. van Eyken, E. C. Thomas, P. J. S. Williams and D. M. Willis proposed three parabolic dish antennas to be envisaged in Longyearbyen. STEL was contacted by the UK group in 1988 and asked for international collaboration with them. The proposal was well received by the EISCAT Council (a private communication from Professor Brekke). A detailed investigation of the scientific and technical case for a polar cap radar was made already by 1990 (Cowley et al., 1990). In September 1989, an EISCAT meeting at Hamburg, which Professor Matuura attended, decided to examine the Polar Cap Radar under consideration of possible collaboration with the Japanese SIR group. We learned later that the EISCAT community was then skeptical to the technical feasibility of the Japanese SIR project with the phased array system. It might be interesting to note that the USA research group represented by the Geophysical Institute of the University of Alaska, considered a plan to install phased array IS radar systems at Poker Flat in Alaska and Resolute Bay in Canada and asked the

Japanese SIR group to join their project about the time. But that proposed collaboration was not considered further, after the proposal of the Polar Cap Radar to the EISCAT. The Japanese SIR project was thereafter discussed as an international project between the STEL and the EISCAT Scientific Association. The "Polar Cap Radar Working Group" was established by the EISCAT Council on May 11, 1990. Their report "The EISCAT Svalbard Radar" (ESR) has 130 pages and lists 7 members of the group (https://eiscat.se/wp-content/uploads/2016/11/1991.pdf). It was submitted to the Council in August 1991. In this

report there was no trace yet of the interest in Japan joining the EISCAT Scientific Association, although this was already an on-going activity. The proposal was formally approved by the EISCAT Council at Uppsala, Sweden, in November 1992.

In June 1990 STEL was established and Professor Oguti was appointed as the first director. The laboratory consisted of four research divisions, Atmospheric Environment, Ionospheric and Magnetospheric Environment, Heliospheric Environment and

Integrated Studies. The division of Ionospheric and Magnetospheric Environment was in charge of the SIR. A unique characteristic of the Laboratory was the function of inter-university collaboration that promoted joint research projects for nationwide research institutions and researchers. The Laboratory was again reorganized in 2015, named "Institute for Space-Earth Environmental Research" that has further expanded its function to clarify the mechanisms and relationships between the Earth, the Sun, and cosmic space, treating them as a seamless system. In December 1990, one of the four major projects of

STEL, the SIR project was proposed (Matuura and Oguti, 1991). In April 1991 at the general assembly of the European Geophysical Society in Wiesbarden Germany, by request of the EISCAT group Professor Matuura on behalf of STEL presented the SIR project of a phased array antenna system at Longyearbyen. The EISCAT group presented their Polar Cap Radar project of the dish antennae system at Longyearbyen. Difference between the two systems with their comparison was reported in Nature News (Aldhous and Swinbanks, 1991).

**2 Japan's Participation in the EISCAT Scientific Association with the Construction of the Second Svalbard IS Antenna**

In August1992 Professor Oguti and Dr. Fujii visited the place of the candidate site of SIR, Ny Ålesund, accompanied by Professor Brekke and Dr. Truls Hansen. At that time STEL still pursued the independent SIR project. After having carefully observed the site, Professor Oguti and Dr. Fujii realized various difficulties in the construction and running of SIR, e.g., rather

severe regulation to reduce any impact on the natural environment that would certainly make the construction of the active antenna difficult.  The following intensive discussion with Brekke can be marked an epoch when Japan seriously started to consider to cooperate with the EISCAT Scientific Association, instead of trying to realize the own independent SIR initiative at Svalbard. On the very next day Professor Oguti already stated to Brekke and the dean of the University of Tromsø to incorporate the Japan's project with the EISCAT proposal. Indeed, in November 1992 Professor Matuura on behalf of STEL

informed Professor Brekke with an official letter that STEL was thinking of changing its future proposal such that Japan would primarily participate in EISCAT Longyearbyen radar plan with Japanese in-kind contribution to the construction of the second

parabolic antenna dish. We later learned that there had been some skepticism in the EISCAT community about how technically the EISCAT membership could be expanded outside Europe. Actually, there were various practical difficulties for Japan's participation in the EISCAT Scientific Association such as how to make large investments in foreign countries retaining ownership, continuous financing through different fiscal years, etc.

We were very fortunate to have two key persons for helping us solve those difficulties during the period of the planning and negotiation with the EISCAT Scientific Association. These two persons were Professor Brekke and Dr. Jürgen Röttger, the Director of the EISCAT Scientific Association of the time. Without the two persons' devoted efforts and without their help, it would never have been possible at all for Japan to join the EISCAT Scientific Association. One of the two persons, Professor Brekke continuously acted in central roles in the collaboration between Japan and the EISCAT community before and during his vice-presidency and presidency of the EISCAT Council. Professor Brekke often visited Japan and stayed at STEL three times as a visiting professor, and tried to persuade the MEXT and Nagoya University for the Japan's participation that would inevitably require a rather large amount of budget at the beginning and stable annual membership fees later. He invited Dr. Nozawa to the University of Tromsø for ten months in 1992 and gave him opportunities of training to operate the EISCAT radars and to process/analyze EISCAT data that were essentially important for the later development of the EISCAT user community in Japan. In 1993 Professor Brekke was elected to chairperson of the EISCAT Council and he accelerated the process for the Japanese participation in the EISCAT Scientific Association. During the General Assembly of the International Union of Radio Science (URSI), which was held at the end of August 1993 in Kyoto, Japan, an exhibition stand "ESR-International Collaboration" was prepared by the STEL of Nagoya University and the EISCAT Scientific Association (The EISCAT Annual Report 1993, https://eiscat.se/wp-content/uploads/2016/06/1993-Annual-Report-scanned.pdf). After the URSI General Assembly in September 1993, the EISCAT delegation consisted of Professor Brekke, Dr. Röttger and Professor Jorma Kangas, the University of Oulu Finland visited Professor Nobuo Kato, the President of Nagoya University and did encourage him to apply for membership of the EISCAT Scientific Association on behalf of Japan. Following this meeting they also visited Mr. Masayuki Inoue, the Director of the Division for International Research at the MEXT and again encouraged him to support the application of Japan becoming a member of EISCAT with stating that EISCAT was considering Japan as the welcomed collaborator and the EISCAT was ready to invite Japan to participate in the EISCAT Council.

In June 1994 STEL asked Mr. Masahiro Nishio, the Director General of Nagoya University to visit Svalbard for an inspection tour of EISCAT. It convinced him of the importance for Japan and Japanese researchers to participate in the EISCAT Scientific Association. After returning to Japan, he immediately started to negotiate with MEXT for the Japan's participation in the EISCAT Svalbard Radar project, which we think triggered the merging of the two projects and Japan becoming an associate of the EISCAT Scientific Association. In late August to early September 1994, the Japan-EISCAT Symposium on the Polar Ionosphere was held in Toba, Japan under co-sponsorship by the STEL and the EISCAT Scientific Association. It was very successful in terms of scientific values and personal ties between European and Japanese scientists, with attendees of 61

scientists including 17 scientists from foreign EISCAT related countries.  Among participants were Mr. Furuya, a high governmental official from the Division for International Research at the MEXT and Mr. Ito, the Director General of Nagoya University.  By this participation  we had a hunch that the STEL's proposal would be approved and funded. Some of the papers submitted to the Symposium were published in the Journal of Geomagnetism and Geoelectricity, Japan (Special Issue. Vol.

47, 1995).   After the Symposium, Professor Brekke, Dr. Eivid Thrane, NDRE Norway, and Dr. Röttger together with Professor Susumu Kokubun, the director of STEL at that time and Professor Matuura visited Mr. Inoue at MEXT again. Mr. Inoue informed them that the MEXT was going to fund for the Japanese EISCAT project to the National Institute of Polar Research (NIPR) instead of Nagoya University. This was probably because the NIPR has been in charge of promoting scientific activities particularly large research projects of Japan in both of the Arctic and Antarctic regions.


In May 1995 funds for the Japan's participation with the in-kind contribution to the second dish antenna were released by MEXT.  The EISCAT Council approved the Japan's participation in the EISCAT Scientific Association as the seventh associate country at the Council Meeting held in Hamburg on 23 May 1995 with concluding the Memorandum of Understanding between the National Institute of Polar Research signed by Director General Takeo Hirasawa, NIPR and the EISCAT Scientific

Association signed by Director Jürgen Röttger. At the same time the EISCAT Agreement among the original six Associate countries ended and the new Agreement was signed by the seven Associate countries including Japan. Delegates and Executives of the Council Meeting were taken in the photo (Plate 1). The first inauguration of the EISCAT Svalbard Radar was held on 22, August 1996 at the radar site in Longyearbyen. Mr. Tadayuki Nonoyama, Japanese ambassador in Norway and delegation from Japan (Plate 2) attended the inauguration ceremony.


This is a major outline of the Japan's participation in the EISCAT Scientific Association. There were, however, certain subjects between Japan and the EISCAT Scientific Association that had to be carefully and comprehensively deliberated, treated and prepared, on which Dr. Röttger, Director of the EISCAT Scientific Association, did play essential and indispensable roles. The Deputy Director Professor Anthony P. van Eyken also helped the process very much particularly from scientific aspects. As

mentioned earlier, the EISCAT Council was sometimes skeptical for the Japan's participation in the EISCAT Scientific Association as an associate country, probably since this was the first case of a new member for the EISCAT and furthermore the participation of a 'non-European' country. Dr. Röttger moved important issues forward with strong leadership, while always having respected the authority, opinions and orders of the EISCAT Council. The issues were for example, preparations for how to deal with a new associate, Japan, in the agreement, i.e., concerning the in-kind contribution for the joining EISCAT,

the right of Japan such as the allocation method of the observation time for Japan on the in-kind and annual contributions, etc. He made, with careful and well-thought-out strategies, his best efforts to provide both of the EISCAT Council and Japan with acceptable, possible proposals, with having observed and examined carefully both parties.

Dr. Röttger and his colleagues at the EISCAT Headquarter in Kiruna intensely supported us in technical and financial

subjects for, e.g., the construction of the second antenna. Soon after the intension of the MEXT to fund the second antenna dispatched from Inoue in September 1994 when the EISCAT delegation had visited him, STEL and NIPR started an investigation of the antenna and contacted Kvaener Kamfab AB that had constructed the first Svalbard IS antenna in order to collect necessary information. The category of the MEXT budget for the second antenna, however, had a difficult restriction that the budget had to be used in principle in one year, at most in two years, although construction works in Svalbard could

be made only in a short period around summer and it was clear that two years were too short for completing the antenna construction. Before and after the MEXT's release of the fund to the second dish antenna in May 1995, under such difficult circumstances. Röttger and his headquarter colleagues helped us very intensively and finally the second antenna was installed in due time, 1999. The second antenna of the EISCAT Svalbard radar was constructed by a French company Alcatel selected in 1998 and in operation 1999. The inauguration of the second antenna was held in May 2000 (Plate 3).


## 3 EISCAT Related Achievement by Japanese Scientists

The Japan's participation made the EISCAT community more global, and Japan has established a trustworthy position in the EISCAT Scientific Association in close collaborations with the EISCAT associate countries. The activity of the EISCAT research community in Japan hosted by NIPR and STEL has been growing year by year. Nationwide researchers and graduate

students have been enjoying the radar experiments at Ramfjordmoen and in Longyearbyen (http://polaris.nipr.ac.jp/~eiscat/en/). Furthermore, we have been conducting more comprehensive and coordinated projects together with simultaneous ground-based and space-borne observations in close collaboration with EISCAT community, where in most cases the central sites of the projects have been located at Ramfjordmoen and in Longyearbyen. For example, Pulsating Aurora Project (PsA) with three stations in northern Scandinavia has been conducted since 2015 (http://www.psa-research.org/english/).


Japanese scientists have conducted researches with EISCAT in a variety of science themes; the lower thermosphere wind dynamics considering energy and momentum inputs from below and above, the Magnetosphere-Ionosphere-Thermosphere coupling, characteristics and driving mechanisms of ion upflow, electrodynamics of current, electric field and particles associated with substorms, characteristics and production mechanisms of auroras such as pulsating aurora and patch aurora,

and aurora tomography. Some of the scientific achievements obtained from these researches are briefly described in the following.

Nozawa and Brekke [1995] showed diurnal amplitude of neutral wind enhanced by factor 3 between quiet and disturbed days, and Nozawa and Brekke [1999a, 1999b] showed seasonal and solar cycle variations of the mean, diurnal and semidiurnal components of the neutral wind between 95 and 120 km in the lower thermosphere. Fujii et al. [1998] showed the neutral wind

mechanical energy transfer rate is comparable to the Joule heating rate in the lower thermosphere. Based on the simultaneous ESR and VHF radar, Ogawa et al. [2000] showed that field-aligned ion upflow observed at 665 km in the dayside cusp were

associated with significant anisotropy of ion temperature, isotropic increases of electron temperature and enhancements of electron density. Ogawa et al. [2010] further found that the upward ion flux is generally high when solar activity is high than it is low. Fujii et al. [2012] proposed a new physical process for the latitudinal motion of an auroral arc based on the four-side bound Cowling channel model. Hosokawa et al. [2016] visualized, for the first time, how the gradient-drift instability (GDI) stirs the patch plasma and such a mixing process makes the trailing edge more gradual. Miyoshi et al. [2015], using simultaneous Arase satellite and ground-based observations, revealed that electrons with a wide energy range simultaneously precipitate into the ionosphere in association with the pulsating aurora, providing the evidence that pulsating auroras are caused by whistler chorus waves. Fukizawa et al. [2022] using aurora observation networks during the Pulsating Aurora Project in northern Scandinavia showed that the horizontal distribution of precipitating electrons associated with PsAs could be effectively reconstructed from ground-based optical observations. Based on long-term variations of plasma temperatures in the polar thermosphere. Ogawa et al. (2014) have made study of the upper atmosphere cooling based on 33 years EISCAT data. Furthermore, Japanese scientists so far launched 8 Japanese rockets from the Andoya Rocket Range (now called Andoya Space) or Ny Ålesund. Using the coordinated observational data during one of these Japanese rocket experiments; the DELTA rocket campaign (Abe et al., 2006; Nozawa et al., 2006), Kurihara et al. (2009) indicated that large vertical winds must be responsible for the fast response of the vertical wind to a heating event.

It is noted that under collaborations with the University of Tromsø (UiT), Japanese scientists have operated/installed a variety of instruments for comprehensive and coordinated observations, such as photometers (Adachi et al., 2017; Nozawa et al., 2018), MF radar (Nozawa et al., 2003), Meteor (MR) radar (Hall et al., 2005), sodium LIDAR (Nozawa et al., 2014), EMCCD TV cameras (Hosokawa et al., 2023), digital camera (Nanjo et al., 2022), FPI (Shiokawa et al., 2012), and all-sky imagers (Ogawa et al., 2020) at Ramfjordmoen. The two meteor radars at Tromsø and Alta, where Japanese scientists are co-owners, are part of Nordic Meteor Radar Cluster (Stober et al., 2021). These instruments have been widely used together with EISCAT radars to understand auroral and polar sciences mentioned above.

The number of published refereed journal papers related to EISCAT by Japanese scientists is 240 (160 by Japanese first authors) as of 2021.

## 4 Concluding Remarks

Twenty-seven years have passed since the Japan's participation and thirty-four years since the first contact of Professor Oguti with Professor Brekke. It may be worth to mention that the Japanese EISCAT activity has been placed as an important component among the very long-term Japan and Europe, particularly with Norway, collaborations. Historically Professor Kristian Birkeland stayed in Tokyo and died there in 1917 and Japan was one of the 14 original signing countries of the

Svalbard Treaty in 1920. The recent collaboration started in the mainland of Norway in 70's between Professor Oguti, the University of Tokyo, and Professor Alv Egeland, the University of Oslo for ground-based observations with magnetometer.

The collaborations in Svalbard started in 1985 for ground-based observations with magnetometer and scanning riometer at Ny Ålesund. Observations/ measurements with rockets and balloons had also been started well before the Japan's participation in the EISCAT Scientific Association. Coordinated network ground-based observations with a variety of instruments mentioned in Chapter 3 have been conducted in Mainland of Norway and in Svalbard and in Sweden and Finland. Furthermore, data taking from a Japanese sun observation satellite, Hinode, took place at Longyearbyen. The observation projects after 2010 in

Svalbard have also been involved in Svalbard Integrated Arctic Earth Observing System (SIOS). These wide range of recent observation activities with ground-based instruments mentioned in Chapter 3 and with 8 rockets and Hinode satellite under international collaborations along with EISCAT mainland and Svalbard radar programs are summarized in Fig. 3. We wonder whether the experience and trust between Japan and Norway earned through these activities lead to STEL's plan for a Svalbard IS Radar project and to the later joining of the EISCAT Scientific Association. Now the EISCAT_3D (McCrea et al., 2015),

whose fast time sampling capability makes it actually 4D, is expected to provide us with new astonishing nature and insights of the space around the earth that no one has ever seen before. The Japanese EISCAT community has been making extensively preparations for the EISCAT_3D being constructed and expanding ground-based observations with great excitement.

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

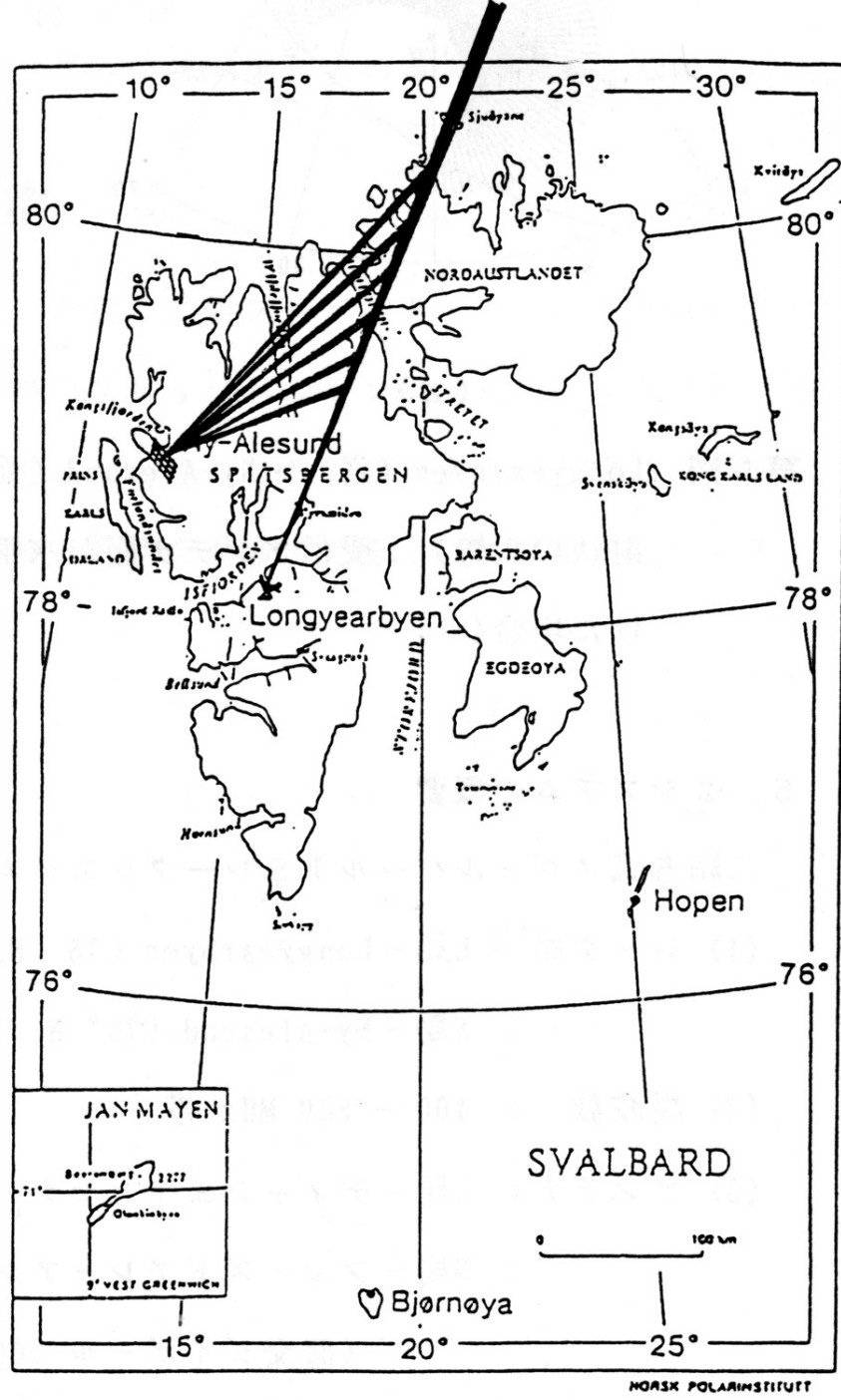

**Figure 1: Bi-static Svalbard IS Radar (SIR) system planned originally by STEL (Matuura and Nozawa, 1991)**

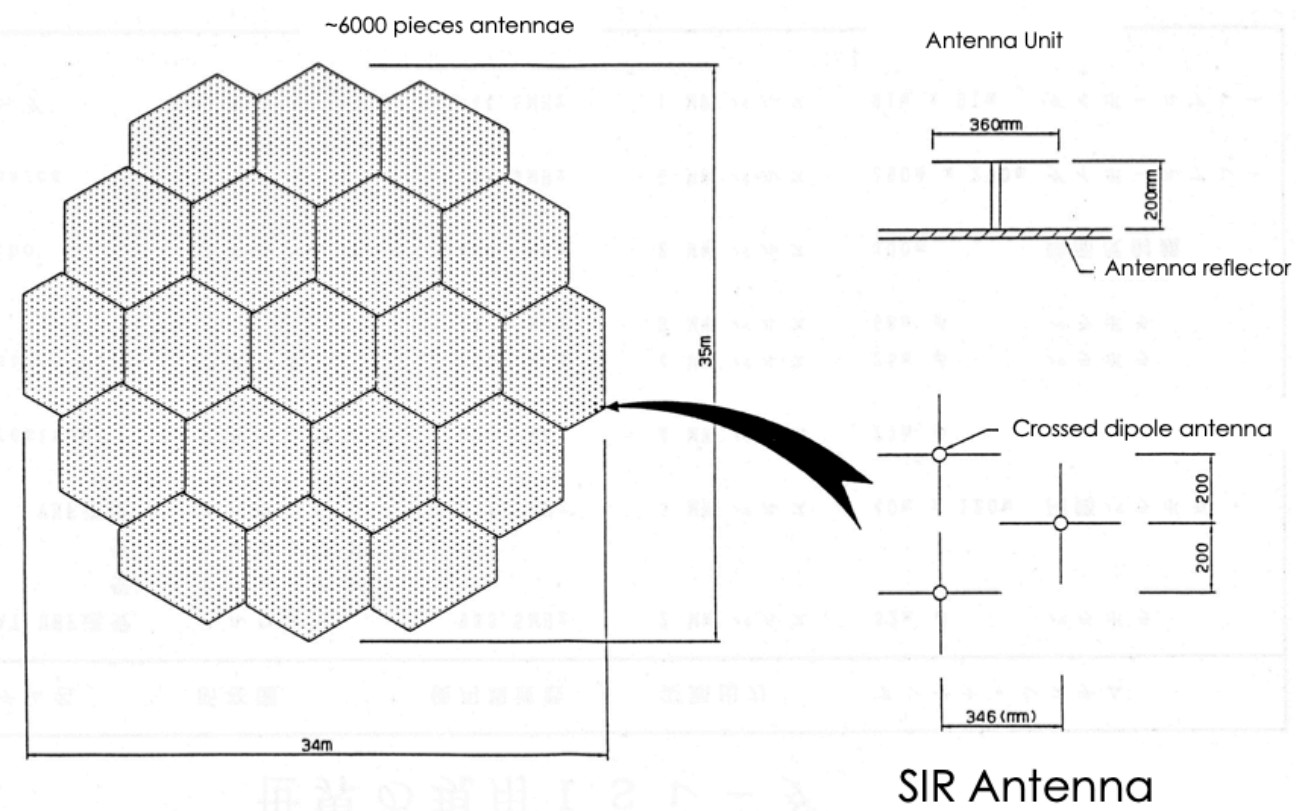


**Figure 2: The structure of the SIR phased array antennae. (Matuura et al., 1990)**

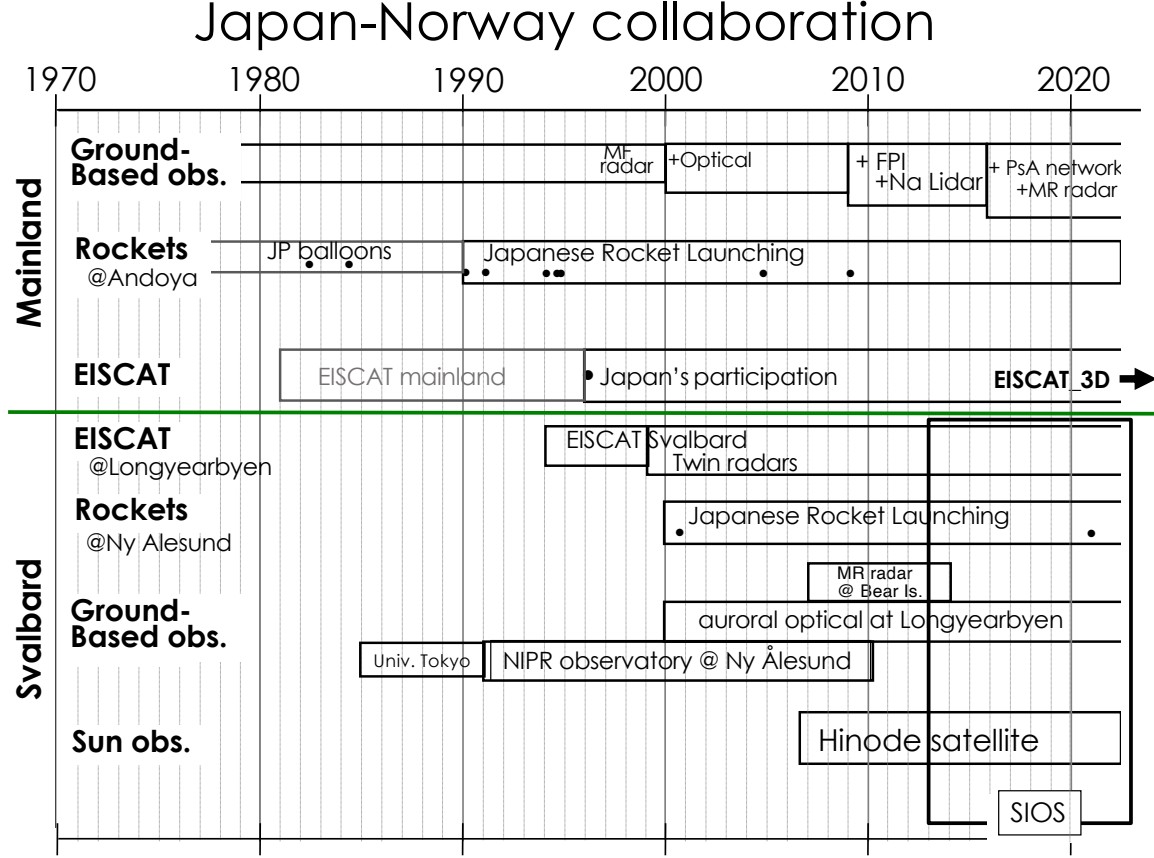

**Figure 3: History of the Japan-Norway/Europe collaborations in space science**

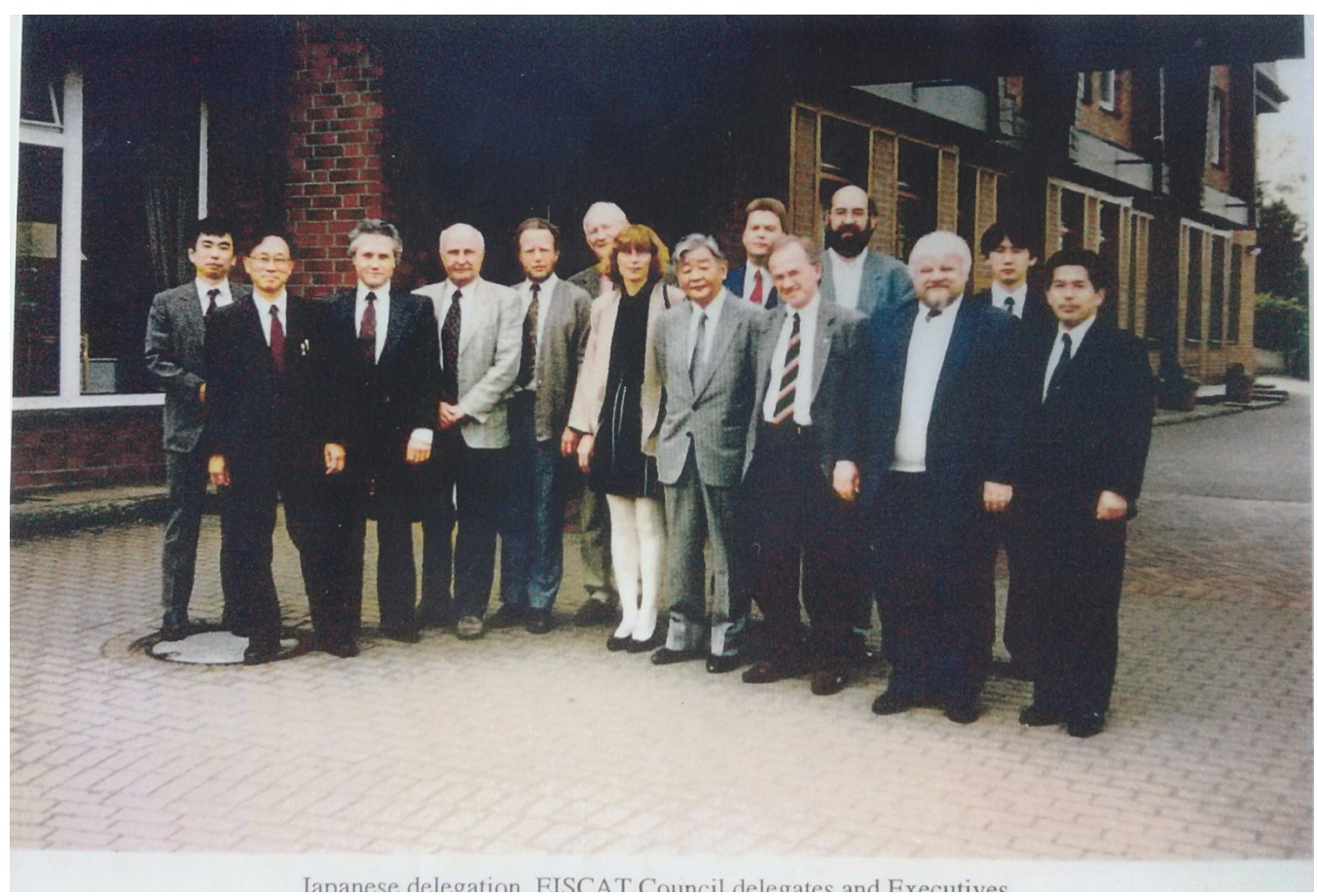

Japanese delegation, EISCAT Council delegates and Executives

**Plate 1: Japanese delegation, EISCAT Council delegates and Executives at the Council meeting on 23 May 1995 in Hamburg.**

 **(Prof. Hirasawa Director of NIPR in the center of the first row, Prof. Brekke on the right, Dr. Röttger the fourth from left,  Prof. Fujii and Prof. Kokubun on the left)**


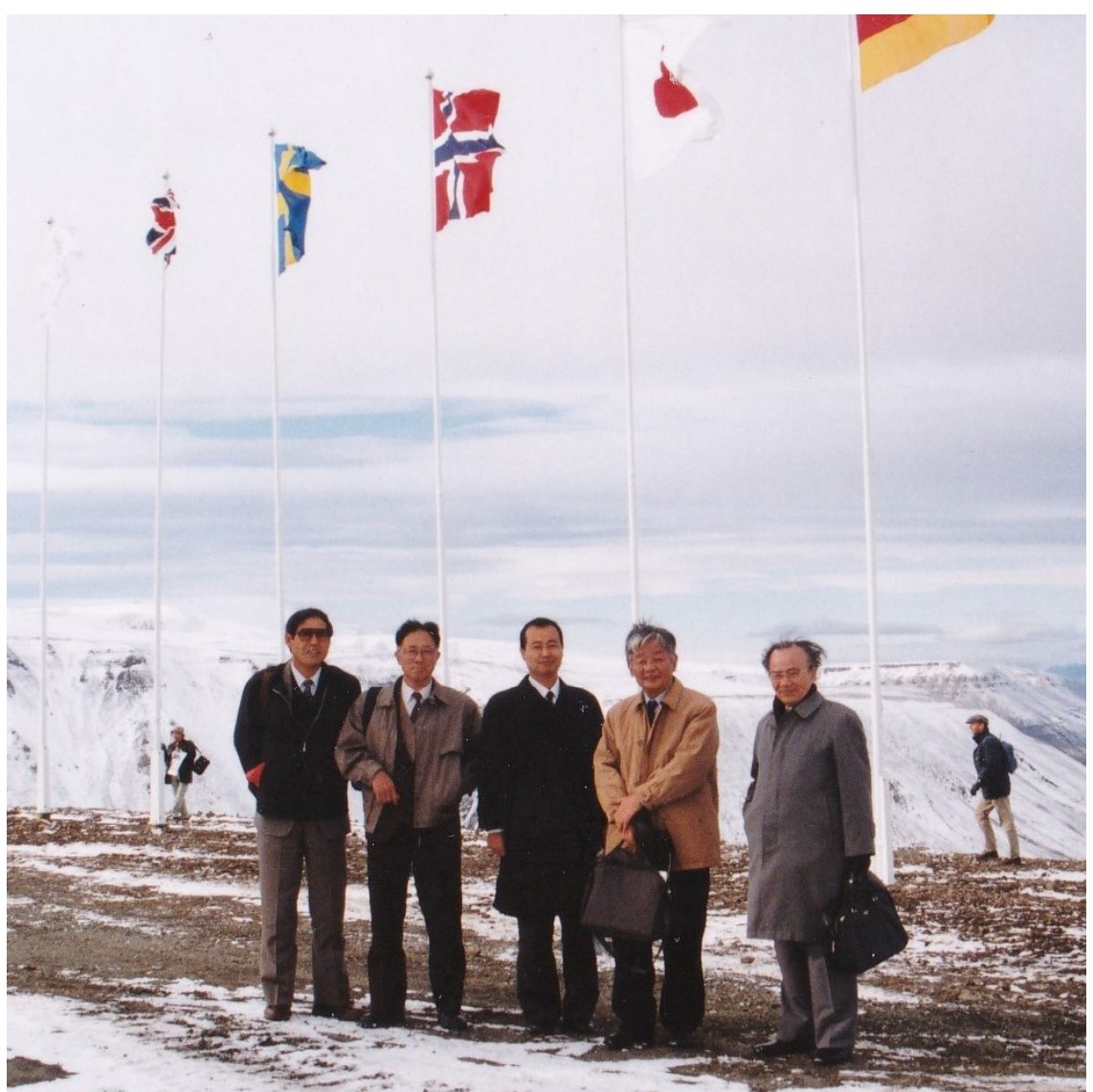

**Plate 2: Japanese delegation at the Inauguration of the EISCAT Svalbard Radar on 22, August 1996 in Longyearbyen**
**(Prof. Matuura on the rightmost, Prof. Hirasawa Director General of NIPR on the second right and Prof. Kokubun Director of STEL on the second left)**


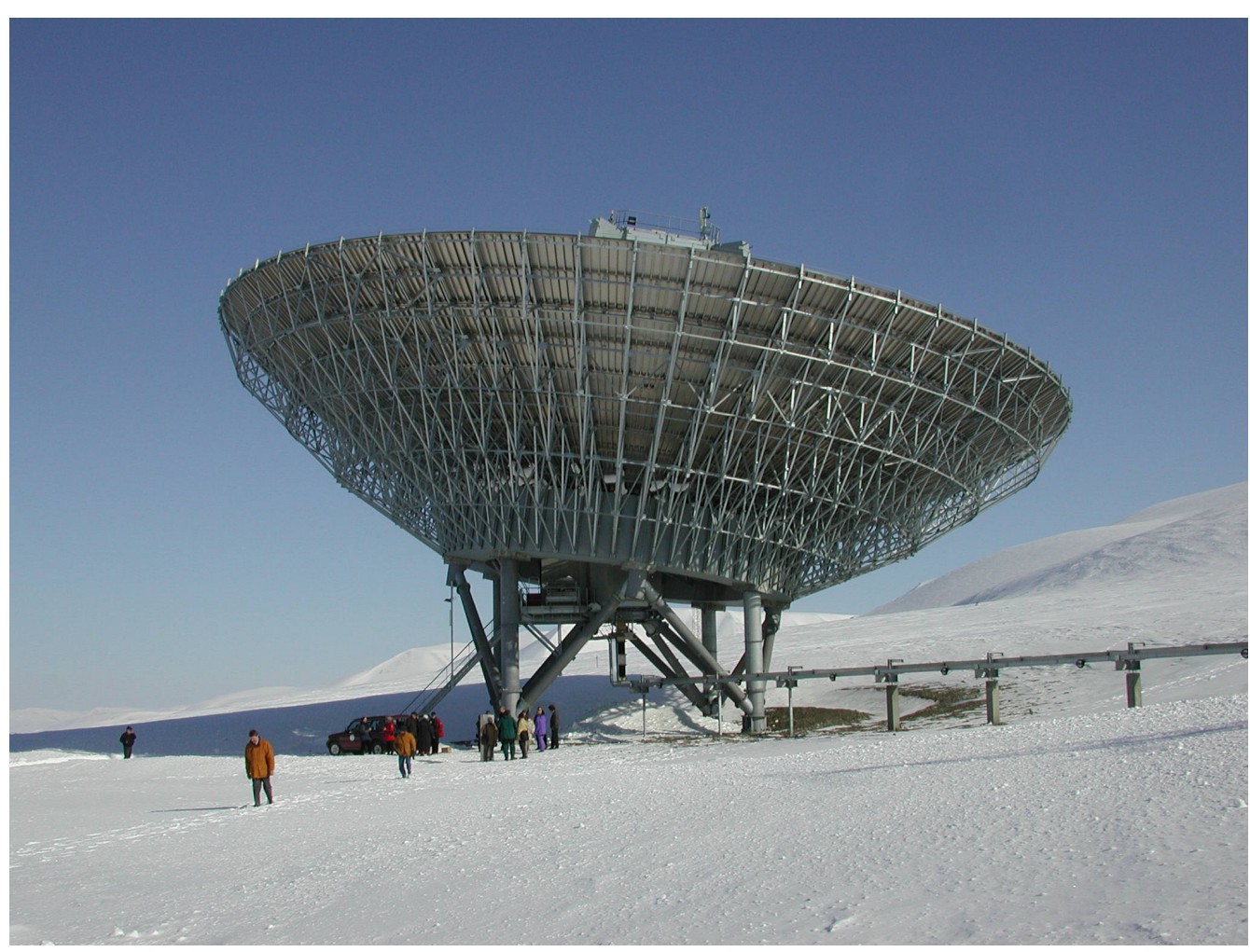

**Plate 3: The Second dish antenna of the EISCAT Svalbard Radar taken on occasion of the inauguration in May 2000**
