# Peer review of "The participation of Japan in the EISCAT Scientific Association"

_History of Geo- and Space Sciences, 2023_

## Author Response (AR1)

Point-by-point reply to the reviewers

We thank the two reviewers, Professor Esa Turunen and Dr. Stephan Buchert, very much for their very thoughtful and valuable comments to our paper. They are extremely helpful and we have revised the manuscript with taking almost all of them accordingly.

The original way of giving the titles of persons was that we referred to the full name with the title when they appeared at the first time and after then only their family names without any titles. We now put titles with the family names all the time through the paper.

We describe below our response one by one to the reviewers' comments, every of which is underlined.

We give serial numbers ((1)~(40)) to the reviewers' comments below and these numbers are referred in the attached marked-up manuscript version showing the changes we have made.

RC1: Professor Turunen's comments

<Necessary minor revisions>

(1) Expand the abstract to better summarize the content of the paper.

 With following the comment above, we have revised the abstract as follows.

**Abstract.** In Chapter 1, the original planning of Japanese Svalbard IS Radar with phased array antennas is described.  In 1988 the plan was proposed as one of major projects for the forthcoming Solar-Terrestrial Environment Laboratory, Nagoya University to be reorganized from the Research Institute of Atmospherics in Nagoya University.  On the other hand, in 1989 UK scientists proposed a plan of the Polar Cap Radar with parabolic dish antennas in Longyearbyen to the EISCAT Council. In Chapter 2, the circumstances arriving in the Japan's participation to the EISCAT Scientific Association are described.  In 1995 Japan participated EISCAT Scientific Association as the seventh member country with funds for contribution to the second dish antenna of the EISCAT Svalbard Radar. In Chapter 3, a summary of the EISCAT related achievement by Japanese scientists is described, where major interests are the lower thermosphere wind dynamics, the Magnetosphere-Ionosphere-Thermosphere coupling, characteristics and driving mechanisms of ion upflow, electrodynamics of current, electric field and particles, characteristics and production mechanisms of auroras such as pulsating aurora, and aurora tomography. In Chapter 4, summary of scientific collaborations between Japan and Europe, particularly, those between Japan and Norway, and hope for the forthcoming EISCAT_3D and further collaboration with EISCAT community are described.

(2)  Expand Chapter 3 to open up the timeline presented in figure 3 as a proper text. Or according to choice of the authors, make it into 2 chapters. Explain all the details of the figure 3 to some extent. Add references where needed. Some reference, summarizing Japan's Scientific Achievements with EISCAT and improving EISCAT's science output, would be of benefit. These are of course many, but what to possibly include, may be judged by the authors themselves.

With following the comment above, we have made the original chapter 3, Concluding Remarks, into two chapters, 3 EISCAT Related Achievement by Japanese Scientists and 4 Concluding Remarks, as follows. Also added are more explanations of Figure 3.

[revised manuscript text omitted]

(3)  I encourage the authors to add original references throughout the paper, in addition to those I mention in the detailed comments, so that the interested reader would be given a possibility to go deeper in the subject.

   With following the comment above, we have included some original documents. Many new references are added with the new chapter 3, 3 EISCAT Related Achievement by Japanese Scientists. The planned new references are as follows.

[revised manuscript text omitted]

Detailed comments:

(4) Lines 8-10: It would be better to use the official international name "EISCAT Scientific Association" throughout the paper

We have revised as advised.

(5) Line 29: Break the sentence after "STEL in 1988" and start a new one "Since the…".

We have revised as advised.

(6) Line 35: Use words "crossed dipole"

We have revised as advised.

(7) Lines 36-38: Reformulate and expand these lines. Add some words here describing the incoherent scatter capability of the Shigaraki MU radar. Remove any parentheses when referring to international projects. Most importantly add a reference, or even two.

We have revised as advised. We have added two references, Fukao et al., 1985a and 1985b. (Please see the list in 3) above.

(8) lines 49-50: Wouldn't it be better to detail a bit, like "skeptical to the technical feasibility of the Japanese SIR project…)

We have revised as advised.

(9) Lines 53-54: This would be better readable, if authors state the last sentence slightly different, for example: "But that proposed collaboration was not considered further, after the proposal of the Polar Cap Radar to the EISCAT.   The SIR project was thereafter discussed as an international project between the STEL and the EISCAT Scientific Association."

We have revised as advised.

(10) Line 62: Remove the parentheses. The sentence forms a very important piece of information for the reader in the paper.

We have revised as advised.

(11) Lines 65-68:   Break the long sentence and remove the parentheses. Important content here.

We have revised as advised.

(12) Line 69: should be "…in Nature News…"

We have revised as advised.

(13) Lines 69-71: The reader is left with a question mark in his/her mind after reading these sentences. There is explanation later (lines 129-136). But more explanation would be needed here when the topic appears for the first time, especially since this is a paper on historical facts. I of course have no knowledge about the difficulties mentioned here. So below you find just an example of possible formulation. Can the authors mention some real facts, so that the reader can use some logic while reading. Maybe one could state something like for example: "We later learned that at that time there had been some skepticism in the EISCAT community about how technically the EISCAT memberships could be expanded outside Europe. And actually, there were various practical difficulties for Japan's participation in the EISCAT Scientific Association, such as how to make large investments in foreign countries retaining ownership, continuous financing commitments through different fiscal years etc." So this was just an example in order not to leave the reader confused.

We have revised as advised. Thank you for your thoughtful comment.

(14) Lines 73-75: I think this is a very important paragraph in a history paper. I would suggest to highlight this a bit. First: break the sentence on line 74 like "…with EISCAT. These two key persons were…", Second: add title for both persons "Prof. Brekke and Dr, Jürgen Röttger" (with correct spelling). On line 75 write full sentence: "..two person's devoted efforts and without their help it would never…"

We have revised as advised.

(15) Line 76-77: correct to"...acted in central roles in the collaboration between Japan and the EISCAT before and during his vice-presidency and presidency of the EISCAT Council..."

We have revised as advised.

(16) Line 84: Correct to: "…was still sticking to the SIR project…"

We have revised this portion as "At that time STEL still pursued the independent SIR project.", as suggested by Dr. Stephan Buchert.

(17) Line 85: Correct to: "…in the construction and running of SIR…"

We have revised as advised.

(18) Line 86: Here it might help to add a sentence (any reference here?). Most notably, at New Ålesund there was a general plan by Norwegian authorities to develop the whole area as a radio noise-free environment, where a large power active radar transmitter would not really fit in long-term planning.

We have added the sentence "to reduce any impact on the natural environment"

(19) Lines 86-87: I would suggest to reformulate the sentence to be more readable, such as: "…The following intensive discussion with Brekke can be marked as an epoch when Japan seriously started to consider to cooperate with the EISCAT Scientific Association, instead of trying to realize the own independent SIR initiative at Svalbard."

We have revised as advised.

(20) Line 91: add here more words like "…construction of the second parabolic antenna dish."
We have revised as advised.

(21) Line 93: correct to: "…was elected to chairperson…"
We have revised as advised.

(22) Line 106: add full name of the workshop, any reference to the workshop proceedings etc..?
   We have added the following sentences with following the suggestion.
"the Japan-EISCAT Symposium on the Polar Ionosphere was held in Toba, Japan under co-sponsorship by the STEL and the EISCAT Scientific Association."
"Some of the papers submitted to the Symposium were published in the Journal of Geomagnetism and Geoelectricity, Japan (Special Issue. Vol. 47, 1995). "
"Special Issue of the Japan-EISCAT Symposium, Edited by N. Matuura and Y. Kamide, Journal of Geomagnetism and Geoelectricity. Vol. 47, Nos. 8 and 9, 1995" is listed in the reference.

(23) Line 108: correct for example to: " Among participants were…"
We have revised as advised.

(24) Line 110: Break the sentence: "…University. By this participation we…"
We have revised as advised.

(25) Line 113: NIPR is mentioned for the first time in paper here. Please, can you add some more description of the institute here, its relation to STEL and maybe why the financing was directed in this way? If there is reference to NIPR history, it would be good.

We have added the following sentences with following the suggestion.

"Mr. Inoue informed them that the MEXT was going to fund for the Japanese EISCAT project to the National Institute of Polar Research (NIPR) instead of Nagoya University. This was probably because the NIPR has been in charge of promoting scientific activities particularly large research projects of Japan in both of the Arctic and Antarctic regions. "

(26) Line 118: The original reference does not list people in Plate 1. Maybe the Authors could pinpoint some of the people mentioned in the paper here, or in the caption? Is Director General Hirasawa in the center, first row? Brekke on his right? Röttger 4th from left? Two Japanese delegates on left, Prof. Fujii and Prof. Kokubun?

We have revised as advised.

(27) Line 121-122:   There is erroneous text here, which needs to be corrected. I think the shown picture of the second antenna of the EISCAT Svalbard Radar is taken during the inauguration of the 2nd antenna, which happened in May 2000. This could be added in text or figure caption, when describing the picture. But even more importantly the paragraph containing these lines should be corrected by adding some sentences. The whole paragraph mainly speaks about joining the EISCAT Scientific Association, which officially happened during the same year as the EISCAT Svalbard radar, with one steerable antenna, was inaugurated, which is 1996. The second antenna was constructed as the next project of the EISCAT Scientific Association on Svalbard. And it was made possible by Japan now financing it since 1996. And the second antenna was inaugurated in 2000 (it's use started in 1999).

Thank you for pointing out our misunderstanding/mistakes. We have revised the portions accordingly like

"Delegates and Executives of the Council Meeting were taken in the photo (Plate 1). The first inauguration of the EISCAT Svalbard Radar was held on 22, August 1996 at the radar site in Longyearbyen.   Mr. Tadayuki Nonoyama, Japanese ambassador in Norway and delegation from Japan (Plate 2) attended the inauguration ceremony."

"The second antenna of the EISCAT Svalbard radar was constructed by a French company Alcatel selected in 1998 and in operation 1999. The inauguration of the second antenna was held in May 2000 (Plate 3)."

(28) Maybe the timeline of 2nd antenna construction would need some sentences here, or somewhere else. I would encourage the authors themselves to formulate how to summarize at least some of the following facts: As timeline in general, the EISCAT Scientifc Association took decision to develop ESR in 1990, with formal decision on construction in 1992. First measurements were

made on 16 March 1996 and ESR was ceremonially inaugurated on 22 Aug 1996. Installation of the second dish was made possible by Japan joining the EISCAT Scientific Association in 1996 and committing to finance the second antenna. In 1997 EISCAT evaluated industrial proposals for the 2nd antenna and finally selected the one by a French company Alcatel in January 1998. The construction was smooth, finished in schedule with first light of the 2nd antenna in autumn 1999, and official inauguration was held in May 2000.

We have mentioned all these facts in Chapters 1 and 2.

(29) lines 133-134: remove parentheses, by adding some words to bind the sentences. Or it would be better to break the sentence to 3 separate sentences.

We have revised as advised.

"The issues were for example, preparations for how to deal with a new associate, Japan, in the agreement, i.e., concerning the in-kind contribution for the joining EISCAT, the right of Japan such as the allocation method of the observation time for Japan on the in-kind and annual contributions, etc."

(30) Lines 138-146: A company is mentioned here, as regards the construction of the 1st ESR antenna. A reference to this antenna should be added (any document or publication describing the antenna). The company selected for the 2nd antenna by EISCAT (where Japan already was a decision making member) should also be mentioned, as well as a reference for the antenna similarily as for the 1stantenna.

We have revised as advised.

(31) Line 146: timeline reference should be added here, ie. what was due time.

We have added 1999 as the due time like:

"Röttger and his headquarter colleagues helped us very intensively and finally the second antenna was installed in due time, 1999. The second antenna of the EISCAT Svalbard radar was constructed by a French company Alcatel selected in 1998 and in operation 1999. The inauguration of the second antenna was held in May 2000 (Plate 3). "

(32) Lines 148-162: The two paragraphs should be rewritten, as mentioned in my minor revision request number 2. I encourage the authors to include some more words in summary (includig possible references) on Japan's scientific achievements using EISCAT and how Japanese participation improved the science output of EISCAT.

As mentioned earlier, we have made a new chapter 3 for Japanese scientific achievements.

RC2: Dr. Stephan Buchert' comments

(33)The UK report "The Polar Cap Radar" is signed by four authors in alphabetical order: A. P. van Eyken, E. C. Thomas, P. J. S. Williams, D. M. Willis

We have referred to the UK report in the manuscript as

"On the other hand, the UK report "The Polar Cap Radar" signed by A. P. van Eyken, E. C. Thomas, P, J. S. William and  D. M. Willis proposed three parabolic dish antennas to be envisaged in Longyearbyen."

(34)Line 84: "... STEL was still stick to the SIR project independent of EISCAT." ---> "... STEL still pursued the independent SIR project."

We have revised as advised.

(35)Lines 85-86: "... rather severe regulation to the environment ... " ---> "... rather severe regulation to reduce any impact on the natural environment ..."

We have revised as advised.

(36)Line 149: "It may be worthy to mention ..." ---> "It may be worth to mention ... "

We have revised as advised.

(37)Lines 155-157: "We wonder that the experience and trust between Japan and Norway earned through these activities let the newly established STEL plan the Svalbard IS Radar project and later join the EISCAT Association." ---> "We wonder whether the experience and trust between Japan and Norway earned through these activities lead to STEL's plan for a Svalbard IS Radar project and to the later joining of the EISCAT Association.

We have revised e as advised.

(38)The "Polar Cap Radar Working Group" was established by the EISCAT Council on May 11, 1990. Their report "The EISCAT Svalbard Radar" has 130 pages and lists 7 members of the group. It was submitted to the Council in August 1991. A scanned version is available at https://eiscat.se/wp-content/uploads/2016/11/1991.pdf

We have used this description in the revised manuscript.

(39) In this report there is no trace yet of the interest for Japan joining EISCAT, although this was certainly already an on-going activity. This perhaps is an indication of the initial scepticism about a non-European Japan in EISCAT, which is also described in the manuscript.

We have also used this description in the revised manuscript.

(40) The EISCAT Annual Report 1993, available at https://eiscat.se/wp-content/uploads/2016/06/1993-Annual-Report-scanned.pdf has an extended "Summary of the Year", authored again by Jürgen Röttger. Here now contacts between EISCAT and STEL, Nagoya University, are officially mentioned, as well as an exhibition "ESR - International Collaboration" at the General Assembly of the URSI in Kyoto. The visit by the Council chairman, A. Brekke, Prof. Kangas from Finnland and the EISCAT director J. Röttger to Nagoya University and their reception by the president and then also by the Director for International Affairs at the Japanese Ministry for Education (Monbusho) are described as they are also in the manuscript.

We have added the following description.

During the General Assembly of the International Union of Radio Science (URSI), which was held at the end of August 1993 in Kyoto, Japan, an exhibition stand "ESR-International Collaboration" was prepared by the STEL of Nagoya University and the EISCAT Scientific Association (The EISCAT Annual Report 1993, https://eiscat.se/wp-content/uploads/2016/06/1993-Annual-Report-scanned.pdf).